# Smart Polymeric Micelles for Anticancer Hydrophobic Drugs

**DOI:** 10.3390/cancers15010004

**Published:** 2022-12-20

**Authors:** Andy Guzmán Rodríguez, Marquiza Sablón Carrazana, Chrislayne Rodríguez Tanty, Martijn J. A. Malessy, Gastón Fuentes, Luis J. Cruz

**Affiliations:** 1Translational Nanomedicine and Imaging Group, Department of Radiology, K2-24g room, Leiden University Medical Center, Albinusdreef 2, 2333 ZA Leiden, The Netherlands; 2Cuban Center of Neurosciences, La Habana 11600, Cuba; 3Department of Neurosurgery, J-11-R-84 room, Leiden University Medical Center, Albinusdreef 2, 2300 RC Leiden, The Netherlands; 4Biomaterials Center, University of Havana, La Habana 10400, Cuba

**Keywords:** smart polymeric micelles, anticancer hydrophobic drugs, nanocarriers, smart drug-delivery systems, gold nanoparticles

## Abstract

**Simple Summary:**

Cancer has been a lethal and high-incidence disease for many years and its cure is characterized by invasive methods and with notable side effects. Nanotechnology has come to shorten the gap in the search for the holy grail of a cure for cancer. This work aims to contribute, from the exposure of the different nanocarriers that can be used as vehicles for the transport of anticancer drugs, to the knowledge of the new routes to reach the final goal, which is none other than to reduce the incidence of this lethal disease to very low levels in humans.

**Abstract:**

Cancer has become one of the deadliest diseases in our society. Surgery accompanied by subsequent chemotherapy is the treatment most used to prolong or save the patient’s life. Still, it carries secondary risks such as infections and thrombosis and causes cytotoxic effects in healthy tissues. Using nanocarriers such as smart polymer micelles is a promising alternative to avoid or minimize these problems. These nanostructured systems will be able to encapsulate hydrophilic and hydrophobic drugs through modified copolymers with various functional groups such as carboxyls, amines, hydroxyls, etc. The release of the drug occurs due to the structural degradation of these copolymers when they are subjected to endogenous (pH, redox reactions, and enzymatic activity) and exogenous (temperature, ultrasound, light, magnetic and electric field) stimuli. We did a systematic review of the efficacy of smart polymeric micelles as nanocarriers for anticancer drugs (doxorubicin, paclitaxel, docetaxel, lapatinib, cisplatin, adriamycin, and curcumin). For this reason, we evaluate the influence of the synthesis methods and the physicochemical properties of these systems that subsequently allow an effective encapsulation and release of the drug. On the other hand, we demonstrate how computational chemistry will enable us to guide and optimize the design of these micelles to carry out better experimental work.

## 1. Introduction

Cancer is one of the leading causes of death worldwide. Surgery accompanied by subsequent chemotherapy is the most widely used treatment to lengthen or save the patient’s life. Surgery inherently bears secondary risks, such as infection and thrombosis. Chemotherapeutic compounds may be ineffective due to low water solubility, low tumor targeting, and low cellular uptake, and they may cause cytotoxic effects on healthy tissues [1]. In cancer treatment, nanocarrier systems have great potential to help or replace traditional chemotherapy. Controlled drug delivery by nanoparticles is crucial but has always been a significant challenge in developing new effective cancer therapies and diagnostics. The nanocarriers most used for these purposes are carbon nanotubes [2], dendrimer [3], micelles [4], quantum dot [5], liposomes [6], cubosomes [7], magnetic nanoparticles [8], gold nanoparticles, and mesoporous silica nanoparticles (Figure 1) [9,10].

These systems, also known as smart drug-delivery systems (SDDS), are developed so that affected cells, but not healthy cells, locally deliver therapeutic agents at the right concentration. The release of these agents occurs by physicochemical mechanisms that depend on external or internal stimuli that act on the nanoparticles [11,12]. These stimuli include pH gradients [13], redox reactions [14], enzymatic cleaving [15], temperature [16,17,18], ultrasound waves [19], light [20], and magnetic and electric fields [8,21].

All these nanocarriers, in one way or another, always contain a hydrophilic polymer to (i) improve their solubility or stability in extracellular fluids, (ii) avoid processes of exclusion by the immune system and rapid elimination by the reticuloendothelial system, and (iii) functionalize molecules (peptides, proteins, antibodies, genetic material, etc.) that allow the drug to be guided to the target and to be released. This conjugation is always done by forming amide, disulfide, or ester bonds [10]. Within SDSS, polymeric micelles have gained renewed interest due to their physicochemical properties that optimize accumulation in tissues caused by the increased vascular permeability in sites of cancer or inflammation, the so-called enhanced permeation, and the retention (EPR) effect that regulates tumor targeting. Furthermore, polymeric micelles notably increase the solubility of hydrophobic drugs in water by 10- to 5000-fold. The drugs are stabilized in a core composed of a hydrophobic polymeric or lipid matrix covered by a hydrophilic polymer. This modification will prolong the circulation of drugs in the bloodstream. The synthesis of micelles as an SDDS constitutes a potential research topic because of the wide variety of biocompatible and biodegradable polymers and lipids [22,23].

This work presents the most recent advances in smart polymeric micelles sensitive to both endogenous and exogenous stimuli that allow for efficient drug delivery. The review constitutes the basis for the development of the doctoral theses of our research group. The study of these topics will allow for the design of new intelligent micelles for the controlled release of all kinds of hydrophobic drugs. The nanocarriers will be capable of loading anticancer, anti-inflammatory, analgesic, and stimulant drugs, diversifying the applications of the micelles. The study was carried out through a systematic bibliographic review in databases such as Web of Science, SCOPUS, and Google Scholar over the last 10 years. The search for information was carried out using keywords or phrases such as “nanocarriers”, “polymeric micelles”, “drug delivery”, “micelles”, “anticancer drugs”, “critical micellar concentration” and “endogenous and exogenous stimuli.”

### 1.1. Polymeric Micelles

Polymeric micelles are formulated by self-assembly in an aqueous medium of block copolymers that spontaneously form a core–shell structure with a marked amphiphilic character. The differences between the hydrophobic and hydrophilic segments will directly influence the micellization process for forming this type of nanostructures. Copolymers are classified according to the number of monomers in their structure, such as di-, tri-, tetra-, and multiblock copolymers.

Polymeric micelles generally have a hydrophobic core with a hydrophilic coating. Most therapeutic cancer agents are hydrophobic and can therefore be encapsulated in the core. These copolymers can be modified unlimitedly, making them either more hydrophobic or hydrophilic, depending on the chemical properties of the drug under study. In this way, the stability and solubility of these drugs in biological systems can be improved [24]. Figure 2 shows the most frequently used polymers as a hydrophobic core.

The core is covered by a layer generally composed of low toxic hydrophilic polymers, stabilizing the system in an aqueous medium. The properties of the cover layer determine how efficiently micelles can circulate in the bloodstream. Figure 3 shows some of the most commonly used polymers [31].

The formation of micelles depends on three parameters: ionic strength, critical micellar temperature (CMT), and critical micellar concentration (CMC). Ionic strength is relevant in most cases when working with ionic surfactants (whose polar group has a charge). The critical micellar temperature (CMT) is nothing more than that temperature above which the formation of micelles is favored. The critical micellar concentration (CMC) is defined as the concentration of surfactants above which micelles are formed. 

The formation of the micelles will depend on attractive forces that lead to the association of molecules and repulsive forces that control the growth of the nanostructure. CMC depends on the block copolymers’ hydrophobic–hydrophilic balance, the polymers’ molecular weight, and their chemical properties [36]. Micelle formation occurs by removing hydrophobic parts from the aqueous medium and re-establishing hydrogen bonds in water (Figure 4). 

These bonds cause a decrease in the interfacial free energy of the system and a greater stabilization of the core. The micellization process is expressed in terms of free energy, CMT, and CMC (Equation (1)) as: ΔG_0_ = RT ln(CMC)(1)
where ΔG_0_ is the standard free-energy change of micellization, R is the gas constant, and T is the absolute temperature of the system [38].

Using amphiphilic copolymers with a low CMC value ensures that polymeric micelles become more stable in blood than liposomes and other surface-active micelles. In some cases, the formation of polymeric micelles is temperature dependent. The temperature of the formulation must always be higher than the critical micellar temperature (CMT), which depends on the structures and properties of the copolymers [39]. The number of surfactant monomers that form a micelle is defined as the micellar association number, which is thermodynamically defined and varies from 10 to 200 monomers [40].

### 1.2. Micelle Synthesis

Five techniques are used to synthesize micelles. These are (1) direct solution, (2) oil in water emulsification, (3) thin-film hydration/solvent evaporation (4) dialysis, and (5) freeze-drying (Table 1). The applied technique depends on the solubility of the selected copolymer. The polymeric micelle is classified as regular or direct if the copolymer’s hydrophilic part is exposed on its surface. On the other hand, the micelles are called inverse when they do not have a copolymer with the hydrophilic part exposed on the surface [10].

### 1.3. Size

The classification of micelles by their size has been the subject of debate. Some authors, such as Qi et al., support the argument that micelles smaller than 30–50 nm are classified as small, simple micelles. Micelles larger than 50 nm are classified as large micellar complexes. As in other nanostructured systems, small micelles generally experiment with a single-step aggregation process during their formation. However, the aggregation process is usually accompanied by several steps in the case of large and complex micelles [54]. The size of a specific micelle depends on the compacting of the copolymer chains. The lengths, the relationship between the hydrophobic and hydrophilic blocks, and the molecular weight of the surfactants will determine the micellar association number and, therefore, the total size of the micelle. 

Crothers et al. demonstrated experimentally how the values of the mass-average association numbers of the micelles (N_w_) determine the shape of the micelle nucleus and its size. If N_w_ reaches high values, micelles tend to form a rod or worm shape [55]. The size of the charged micelles or ionic micelles can be modified by adding salts such as potassium bromide and sodium salicylate. These salts can neutralize the charges on the micellar surface, promoting a pronounced growth of the micelles [56]. The particle size has an important effect on the immune system. Small micelles have high blood circulation and do not cause many activations of the immune system. If their size is less than 20 nm, the renal system filters them rapidly out of circulation before they reach their goal. Generally, it is recommended that the hydrodynamic diameter be between 60 to 150 nm to obtain effective drug delivery and release [57].

### 1.4. Surface Charge

Most micelles used in drug delivery are in the colloidal state and will therefore have a surface charge (positive, negative, or neutral). The surface charge of these nanocarriers can be changed by chemically modifying their surface with hydrophilic polymers, amino acids, or peptides. The dynamic light-scattering technique is used to determine the surface charge of micelles by Z-potential measurements. The higher the value of the modular Z-potential, the better the particles’ colloidal stability and, thus, a higher useful life [58]. Surface charge is undoubtedly important in determining protein adsorption and cellular interactions. In vitro studies show improved cellular uptake and circulation time for charged micelles. Neutral micelles have a longer circulation time but also greatly inhibit plasma proteins’ adsorption to the particle surface. Positively charged micelles have better interaction with cell membranes and internalization but could sometimes have a toxic effect on cells. The negative charge has reflected milder toxicity but decreases the cellular absorption of particles [59]. Positively charged nanostructured systems (<150 nm) tend to accumulate in the liver. However, if these systems are negatively charged, their circulation will increase and their accumulation will be more difficult. Particles greater than 150 nm are retained in the spleen [35]. 

Researchers must reach a consensus between cellular absorption, circulation times, and toxicity of micelles when their surface charge is modified. Kalinova et al. prepared new curcumin-loaded micelles with tunable surface charges. Copolymers were designed using poly(D,L-lactide) and poly((2-dimethylamino)ethyl methacrylate) and poly(oligo(ethylene glycol)methyl ether methacrylate). The charges varied from strongly positive to neutral as the concentration of hydrophilic polymer was increased. The release profiles show that the curcumin was released much faster for the neutral micelles, but these had lower colloidal stability than the positively charged micelles [60]. Xiao et al. prepared micelles loaded with paclitaxel through the self-assembly of new copolymers called telodendrimers, which are formed by polyethylene glycol (PEG) and cholic dendritic acids (CA). The surface-charge effects were studied by conjugating anionic D-aspartic acids (D) or cationic D-lysines (K) at the ends of the PEG chain. Macrophages do not specifically take up micelles with a high negative and positive surface charge. However, micelles with less negative surface charge showed preferential uptake at the tumor site. The slightly negative charge was also able to prevent the rapid removal of micelles from the bloodstream [61].

### 1.5. Shape

The morphology of block copolymer micelles in dilute solution is generally spherical. The lower-energy spherical conformation can minimize the hydrophobic head’s exposure to the bulk aqueous phase. Different morphologies can be obtained by a remarkable increase in the concentration of surfactants, or the reaction temperature increases excessively. Sphere, rod, and star micelles have been accepted as a function of the length of the hydrophobic/hydrophilic blocks, the concentration of surfactants, and the solvent used. 

Kimura et al. prepared different micelles with spherical and rod morphology by varying chain length and degree of polymerization (DP). Homopolymers with a lower degree of DP formed spherical micelles through the multi-chain assembly, whereas homopolymers that were longer than a threshold DP formed rod micelles [37]. Star-shaped micelles have gained particular interest because the polymeric arms provide a structure with excellent solubility and functionality to guide the drug into a tumor environment [39]. Kang et al. created a new pH-sensitive micellar system that reversibly changes its morphology from spherical to worms. The authors reacted cetyltrimethylammonium bromide (CTAB), 4-hydroxybenzaldehyde (HB), and p-toluidine (MB) by a dynamic covalent-bonding hydrotropic mechanism. If the pH increased, the viscosity of the solution first decreased and then increased rapidly. The microscopic micellar aggregates changed from a spherical morphology to a worm morphology. This morphological change is reversible because when the pH of the medium decreases, the particles become spherical again [62].

Li Deng et al. synthesized an amphiphilic copolymer based on poly (ε-caprolactone) (PCL) and hydrophilic P (2-(2-methoxyethoxy) ethyl methacrylate and oligo (ethylene glycol) methacrylate (MEO_2_MA-co-OEGMA)). Micelles were thermo-magnetically sensitive, presenting a lower critical solution temperature (LCST) around 43.5 °C with potential applications in magnetic hyperthermia and the release of drugs facilitated by magnetothermia [63]. Shang et al. prepared another system with a four-armed star morphology. Each star tip was covered by a folic-acid (FA) molecule, which gave the micellar structure greater internalization and more effective active targeting [64].

## 2. Endogenous Stimuli

### 2.1. pH-Responsive Polymeric Micelles

The extracellular pH of normal tissues and blood is 7.4, which tends to be lower in tumor tissues. The pH of most extracellular tumors has values between pH 6.5 and 7.2. However, the pH can be even down at the intracellular level. The reported values range from 5.0 to 6.0 in endosomes and 4.0 to 5.0 in lysosomes. For this reason, the acidity in the extracellular and intracellular compartments constitutes an essential signal for targeting [1]. Most of the bond’s present in a copolymer are sensitive to pH below 6, so the drug will be released in the intracellular environment. The pH sensitivity of micelles has been investigated by determining the values of the basic dissociation constant (pK_b_), the change in hydrodynamic diameter, and the zeta potential of the particles under different pH conditions [65]. pH-sensitive micelles hydrolyze or adapt a mechanism based on protonation and deprotonation of proton-donor groups. This way, drug release will occur in tumor environments with an acidic pH [13]. The release process occurs in many cases through a bond rupture between the hydrophobic and hydrophilic part, which is sensitive to acidic pH formed by connections. Such bonds are hydrazone [66,67,68], acetal [69,70,71], oxime [72], imine [62], benzoic imine [73], ortho-esters, and vinyl ester bonds [74,75]. Sometimes copolymers are produced with hydrolysable side chains such as polysialic acid (PSA) [76]. Table 2 shows examples of pH-sensitive chemical bonds and their degradation products.

Protonation and deprotonation mechanisms will occur in copolymers containing proton-donating groups such as histidine (His), tertiary amino groups, carboxyl groups, and pyridine. In these cases, some hydrophobic copolymers, in principle, become hydrophilic thanks to protonation in an acidic medium, causing the micelle to break and release the drug. Weng et al. used poly (acrylic acid) (PAA) as the hydrophilic backbone of a micelle designed to release doxorubicin at acidic pHs. This polymer has carboxyl groups in its chain that are easily protonated in an acidic environment [77]. 2-(N,N-diethylamino) ethyl methacrylate (PDEA) and poly (β-aminoester) (PAE) have been used for these purposes because they present tertiary amino groups that are also sensitive to acidic pH. Polymers with these characteristics are often used to design dual-sensitive micelles [78]. Liao et al. prepared dual pH-sensitive micelles with the ability to release the drug by breaking a hydrazone bond and protonation of PDEA) remnants in an acidic environment. The surface-charge changes from negative to positive when the pH decreases. Charge reversal, in this case, significantly facilitated cell-uptake and anticancer efficacy [79].

### 2.2. Redox-Responsive Polymeric Micelles

Polymeric micelles sensitive to redox reactions will release the drug due to the action of a reducing agent such as glutathione (GHS) or reactive oxygen species (ROS). The GHS concentration is very high in tumor cell nuclei, around four times that of healthy tissues, whereas the ROS concentration is approximately 50–100 μM in the extracellular matrix of a tumor. Active targeting will promote a redox response due to the capture of micelles by tumor cells. Disulfide bonds have proven to be the most sensitive to GHS, so the strategy in these cases is to connect the micelles’ hydrophilic part to the hydrophilic portion. However, diselenide links have also been studied for these purposes in the last decade [80,81,82].

Yihenew et al. prepared a copolymer by forming a redox-sensitive diselenide bond between an amphiphilic polymer, Bi (mPEG-PLGA)-Se_2_ of mPEG-PLGA and 3,3′-diselanediyldipropanoic acid (DSeDPA). The conjugate formed stable micelles with relatively low critical micelle concentrations, allowing for a high loading capacity of doxorubicin [83]. Jun et al. self-assembled the amphiphilic copolymer tetraphenylethene-poly(aspartic acid)-block-poly(2-methacryloyloxyethyl phosphorylcholine) (TPE-SS-PLAsp-b-PMPC) into spherical micelles. The goal was to encapsulate and release doxorubicin hydrochloride. The drug was released by breaking the disulfide bonds formed by poly(aspartic acid) and TPE-block-poly(2-methacryloxyethyl phosphorylcholine). This study used the aggregation-induced emission (AIE) effect of a new class of fluorescent molecules, such as TPEs, to obtain a system capable of monitoring drug release through fluorescence imaging [84].

Yanyan et al. encapsulated doxorubicin hydrochloride in nano micelles of dextran and deoxycholic acid (Dex-SSDCA). In this study, five copolymers were produced, varying the molecular masses of dextran. Polymers with lower CMC were dextran 20 kDa and 40 kDa with the same degree of substitution so that these would present the highest self-assembly efficiency. Dex20k-SSDCA showed the highest encapsulation efficiency due to the small size of the hydrophobic nuclei of its micelles, allowing them to enlarge and completely contain the drugs [85]. 

Another drug that has been highly encapsulated in redox-sensitive micelles is paclitaxel. Sheng et al. synthesized polyethylene glycol-poly (β-benzyl-L-aspartate) (PEG-PBLA)-SS-paclitaxel (PPSP) micelles that combined high-speed dispersion-stirring and dialysis methods. The study demonstrated the drug’s rapid release by these micelles compared with similar copolymer of polyethylene glycol-poly (β-benzyl-L-aspartate) (PEG-PBLA-CC-PTX (PPCP) that did not present disulfide bonds [86].

Xiaoqing et al. presented a strategy to combat multidrug resistance (MDR) of cancer cells. With this objective, micellar systems composed of hyaluronic acid-g-cystamine di-hydrochloride-poly-Ɛ-(benzyloxycarbonyl)-L-lysine (HA-ss-PLLZ) were created to encode paclitaxel (PTX) and lapatinib (APA). This smart nanostructure is rapidly degraded in high-cellular glutathione concentrations (GSH). It can specifically bind to CD44 receptors, leading to selective accumulation at the tumor site and uptake by MCF-7/ADR cells facilitating effective suppression of tumor growth [87].

Yujie et al. took advantage of endogenous albumin as a drug carrier through its interaction with the ABD035 peptide, which is also part of a copolymer with the composition ABD035-PEG-SS-PTX. The micelle-bearing albumin crossed the tumor vascular endothelium through gp60-mediated transcytosis and did bind to cysteine in the stroma accumulating at the tumor site. At this point, the PTX could be released under a high concentration of GSH [88].

Hoang et al. prepared a ROS-sensitive poly (ethylene glycol)-poly(methionine) (PEG-P(Met)) micelle to encapsulate piperlongumin (PL), which is a pro-oxidant drug in cancer cells. These micelles exhibited specific cytotoxicity on MCF-7 human breast-cancer cells due to a considerable increase in the intracellular level of ROS in the cells. PL was delivered in an intracellular environment, and drug release was clearly observed due to activation of ROS by oxidation of thioether groups of P(Met) [89].

Pei et al. designed a copolymer composed of a polyphosphoester (PPE-TK-DOX), doxorubicin (DOX), and a photosensitizer (Ce6). The micelles were able to release DOX due to breaking the thioketal bond, which is sensitive to species (ROS). The circulation of the micelles was followed by fluorescence and magnetic resonance imaging. Light irradiation induced ROS species, and then the drug was released at the target site [90]. 

### 2.3. Dual pH/Redox-Responsive Polymeric Micelles

An alternative approach to improve the efficient and rapid delivery of drugs has been synthesizing micelles sensitive to more than one stimulus. Dual pH/redox-sensitive polymeric micelles have gained great interest. The copolymers of these micelles undergo the breaking of two bonds. One of these bonds is sensitive to pH, and the other to GSH or ROS species. For this reason, the circulation and release of the drug can be more controlled in the intracellular and extracellular space. These actions will depend on the content of GSH or ROS species and the pH in each medium. Generally, the pH-sensitive segments form the polymeric micellar core, and its coating is composed of a redox-sensitive part and a hydrophilic shell such as PEG [91].

Luo et al. prepared two types of micelles that should release doxorubicin. The proportions of the copolymers:(1) poly(ethylene glycol) methyl ether-b-poly(β-aminoesters) copolymer (mPEG-b-PAE) and (2) poly(ethylene glycol)) poly(b-aminoesters) copolymer disulfide grafted with methyl ether (PAE-SS-mPEG)—were varied. The thermodynamic stability for these systems was higher than for micelles self-assembled by a single amphiphilic copolymer. This stability was due to decreased CMC, which increased the hydrophobic polymer segment. The high concentration of PAE increases the sensitivity to the system’s pH because there will be a greater amount of tertiary amine residues that are easily protonated in acidic environments. On the other hand, the release of the drug will also be stimulated by breaking disulfide bonds in the chemical structure of the copolymer [65].

Huang et al. prepared star-shaped micelles using the copolymers poly(tert-butyl methacrylate)-b-poly(2-hydroxyethyl methacrylate)-b-poly(poly (ethylene glycol) methyl ether methacrylate) (4AS-PtBMA-PHEMA-PPEGMA). In these micelles, doxorubicin release below pH = 5 is caused by the weakening of electrostatic interactions or chemical bonds between doxorubicin and polymethacrylic acid (PMAA). The breaking of disulfide bonds and the protonation of carboxylate groups at acidic pH also contribute to the effective and controlled release of the drug as well [92].

Lo et al. made micelles from an SS-poly (methacrylic acid) diblock copolymer (PCL-SS-PMAA) to encapsulate and release paclitaxel (PTX) and cisplatin (CDDP) to treat lung cancer. They showed that the position of the disulfide bond (core, core-shell interface) affects how easily it can be degraded in reducing environments. The disulfide bonds in the central core took longer to break than when placed on the interface [93].

Xin et al. designed polymeric micelles from the self-assembly of amphiphilic poly copolymers. They used (ethylene glycol)-poly(γ-benzyl L-glutamate), followed by a core crosslinking reaction using cysteamine (ethylene glycol)-poly(γ-benzyl L-glutamate), followed by a core crosslinking reaction using cysteamine. The use of cysteamine, in this case, makes it possible to obtain a more compact spherical micelle capable of being stable in solution for a long time. DOX-loaded micelles had a low release rate at pH 7.4 without GSH. However, the release continually increased in response to a more acidic and reducing environment [94].

### 2.4. Enzyme Sensitive

Tumor cells reproduce, which coincides with the production or overexpression of a large number of abnormal enzymes. Generally, the overexpression of protease and phospholipase enzymes has been used to cause chemical rupture of the micelles and, therefore, drug release. These enzymes are able to degrade and break ester or amide links in the structure of copolymers. [95,96].

The most studied proteases are usually metalloproteinases (MMPs) because they are actively expressed during cancer-cell invasion and metastasis in most cancers. Barve et al. designed a micelle by joining two block copolymers (Cholesterol/PEG) containing the peptide PLGVRK sensitive to the enzyme MMP2. The authors encapsulated cabazitaxel to treat prostate cancer, in which this enzyme is highly overexpressed. The micelle released 80% of the drug consistently in 24 h in the presence of MMP2. In the absence of MMP2 only about 10% of cabazitaxel is released. The micelle exhibited high cellular uptake in cancer cells and inhibited tumor growth in mice bearing prostate-cancer xenografts [15].

Wan et al. encapsulated DOX in a micelle of D-α-tocopherol polyethylene glycol succinate 3350 (TPGS3350) conjugated with GPLGVR peptide and Asp-Glu-Val-Asp (DEVD)-functionalized folic acid. DOX release occurred slowly and was controlled in two steps. First of all, GPLGVR was cleaved under the action of the enzyme MMP-9. DOX then began to be released by removing the outer layer of TPGS3350. The rest of the drug was rapidly released by the action of another enzyme (Caspase-3) on the DEVD peptide. The micelles were shown to have higher effective tumor inhibition than unencapsulated DOX on 4T1 cancer cells [97]. 

Upregulation of phospholipases has been a common pathology in many types of cancer. Phospholipase A2 (PLA2) has become especially relevant in this group of enzymes because it is overexpressed 20 times more in cancerous tumors than in healthy tissues. Tagami et al. prepared micelles-lipid nanocapsules composed of phospholipid (DPPC) and PEGylated block copolymer (Poloxamer 188) to encapsulate DOX. The system was tested on the A549 lung cancer-cell line, which abundantly expresses the PLA2 enzyme. DOX was released rapidly (74.2%, 5 min; 85.5%, 10 min) by degradation of the lipid structure at high concentrations of PLA2. The sensitivity of the micelles to PLA2 increased markedly when the concentration of pegylated DPPC was increased in the different formulations [98]. 

## 3. Exogenous Stimuli

### 3.1. Thermo-Responsive

Changing the temperature is one of the most studied factors in drug delivery because heat can potentially be used as an external or internal stimulus to start drug delivery. The temperature is also slightly higher in tumor tissue than in normal tissue. The heat-sensitive polymers used in these micelles are classified according to the lower and upper critical solution temperature (LCST and UCST). All water-soluble polymers exhibit an LCST phase transition related to changes in polymer–polymer and polymer–solvent interactions. The ruptures of these interactions, which are mostly hydrogen bonds, allow the micelle to break down and release the drug following an increase in temperature [99].

Poly (N-isopropylacrylamide) (PNIPAAm) and poly (N-vinylcaprolactam) (PVCL) are the most widely used thermosensitive polymers in drug delivery, as they have a very low LCST of around 32 °C in aqueous solution [100,101,102]. Chung et al. prepared micelles based on diblock copolymers of PNIPAAm with PBMA (poly (butyl methacrylate)) or PSt (polystyrene) encapsulating adriamycin (ADR). PNIPAAm-PBMA released the drug when the temperature exceeded the LCST, which was impossible in PNIPAAm–Pst. This phenomenon is due to the copolymer’s greater structural microrigidity and low micropolarity [103].

Rejinold et al. synthesized biodegradable thermo-responsive chitosan-g-poly(N-vinyl caprolactam), taking advantage of the complexation and biocompatibility capacity of PVCL for encapsulated curcumin. This system showed a critical solution temperature of 38 °C. When the temperature was higher than 38 °C, the density of the particles increased and the drug was released rapidly until it reached a plateau where the release was almost constant and much slower. This fact could be explained by an initial swelling followed by a controlled diffusion [104].

### 3.2. Ultrasound

Ultrasound is widely used in the medical field, with the significant advantage that it is a non-invasive technique. Ultrasound waves can also produce thermal effects by absorption of energy or mechanical products by cavitation. The thermal and mechanical properties can cause micelles to contract, expand, and explode, thereby releasing the drug. However, ultrasound can also improve the permeability of biological barriers, leading to greater cellular absorption of the micelles [105].

Wang et al. presented a new strategy to synthesize polypeptide nanoconjugates (UPDN) covalently linked to DOX. Peptides are made up of amino acids (LHRH-ELP) and were also genetically modified. These compounds respond to ultrasound and pH to release DOX. The authors used a C8 peptide with a cysteine group capable of forming a hydrazone bond. This link is broken in the tumor’s acidic environment accompanied by ultrasound through acoustic cavitation, which allows for greater internalization of the drug [106].

Wu et al. prepared mixed PEO-PPO-PEO triblock micelles (pluronic) loaded with curcumin. These copolymers have been used clinically as FDA-approved pharmaceutical adjuvants, guaranteeing good tissue internalization. The mixed P123/F127 micelles proved good vascular permeability after being exposed to ultrasound at a small load power (2W). After 30 min, approximately 28.34% of the drug had been released. However, the authors showed that the drug-release process is accelerated when a higher loading power is applied. In this case, they applied a power (3W), and 38.64% of the drug was released after 30 min [107]. A similar effect occurred due to transient cavitation on Zhang’s investigation with a PLA-b-PEG micelle at the focal point. Nile Red was irreversibly released as a drug payload model in this case by the chemical breaking of the polymer chains [101,102,108].

### 3.3. Light-Responsive

UV-visible and infrared/NIR light represents a desirable stimulus to release drugs encapsulated in micelles because the light can be regulated by adjusting the wavelength, intensity, and exposure time. None of the variants are invasive, and they present the possibility of being remotely controlled with spatial–temporal resolution [109]. The design of these micelles starts from the conjugation of the copolymers with photochromic groups. These systems are classified into two types: cis-trans isomerization groups (e.g., azobenzene and stilbene) and photoinduced changing groups (e.g., spiropyran, dithienylethene, and diazonaphthoquinone) [110].

Ultraviolet light has been widely used in these fields of nanomedicine. Jora et al. took a solution of pH-sensitive micelles (1,2-dihydroxybenzene and cetyltrimethylammonium bromide) and added diphenyliodonium-2-carboxylate. This compound can generate photoacids when exposed to ultraviolet light and further decreases the pH of the initial solution. The micelles could be applied in drug delivery mediated by a dual stimulus (pH-UV light). Through UV light, we could control the pH at which a future release would occur [20].

Haisheng et al. prepared micelles using light-sensitive pyrene ester linkages through the copolymerization of 1-pyrene-methyl acrylate (PA) and N, N-dimethylacrylamide (DMA). Micelles sensitive to UV light break these ester bonds, changing the morphology from spherical to nanorods and inducing moderate cytotoxicity [111].

Ma et al. prepared micelles of poly (2-methacryloyloxyethyl phosphorylcholine-co-azobenzene methacrylamide) (PMA) dual sensitive to UV and NIR light. They achieved this by encapsulating up-conversion nanoparticles (UCNP), which can convert NIR light excitations into UV/Vis through reversible isomerization transitions of the Azo groups, thus releasing the encapsulated DOX in a controlled way. The DOX release rate was increased by approximately 7% while maintaining a constant pH only for an increase in UV irradiation time over 10 min [112]. The UV light presents low tissue penetration and high phototoxicity, which is why other light sources, such as visible light, should be used. 

Yap et al. solved this problem using visible-light-responsive photo-switchable molecules. In this case, they introduced donor–acceptor Stenhouse adducts (DASA) within the polymeric structure derived from short methacrylate homo-polymers. The authors tested this system by encapsulating ellipticine, which was released efficiently due to the recovery capacity of the copolymers after light irradiation [113]. The near-infrared (NIR) ray is usually more attractive because it can penetrate deeper into tissues. 

Zhang et al. prepared dual redox/NIR light-sensitive micelles. The copolymer was made of a biodegradable photoluminescent polymer with a polycaprolactone disulfide bond (PCL-SS-BPLP) and biotin-polyethylene glycol-cypate (biotin-PEG-cypate). Irradiation with NIR caused the decomposition of cypate and the accelerated release of DOX, which glutathione favored in the cancer cell. Furthermore, NIR irradiation could also rapidly generate reactive oxygen species (ROS) and induce cell death and apoptosis through hyperthermia [51].

### 3.4. Magnetic Field-Responsive

Smart micelles sensitive to a magnetic field are characterized by a core composed of the therapeutic agent and magnetic nanoparticles. Magnetite (Fe_3_O_4_) and maghemite (Fe_2_O_3_) are the most widely used nanoparticles. These nuclei generally have superparamagnetic properties, so they have a high magnetic saturation under the action of an external magnetic field. The micelles can then be guided to a specific target when an external magnetic field is applied. In addition, the nuclear magnetic resonance technique allows high-contrast images of the accumulation of these magnetic nuclei in the body to be obtained and, therefore, of the micelles. When the micelles are in the desired target site, the anticancer drugs will be released by some of the endogenous or exogenous factors discussed above. On the other hand, the therapeutic activity of the drugs could be complemented with the application of magnetic hyperthermia. This technique is capable of causing cell death through electromagnetic waves of 40 °C to 44 °C produced by an external magnetic field on the magnetic nanoparticles [114].

Amphipathic chitosan derivatives have gained recent interest. These polymers have high bioactivity and are easily modified to allow for slow release. Chu et al. synthesized pH-sensitive magnetic micelles by modifying chitosan with hydrophobic alkyl and hydrophilic groups of quaternary ammonium and PEG. The release rate of paclitaxel and superparamagnetic nanoparticles increased with the system hydrophobicity. This phenomenon is due to defects during micelle swelling and can be counteracted by coating a hydrophilic polymer such as polyethylene glycol [115].

Mousavi et al. prepared systems sensitive to redox changes. The micelles used polycaprolactone attached to polyethylene glycol using disulfide bridges. Magnetic saturation of encapsulated super para-magnetic iron-oxide nanoparticles (SPIONs) decreased compared to bare SPIONs [116]. This fact is mainly due to the high concentration of polycaprolactone and the molecular weight of polymers. 

Paclitaxel has also been delivered via magnetic-thermo-responsive micelles. Pourjavadi et al. prepared an amphiphilic triblock copolymer for free-radical polymerization. They used poly (N-isopropyl acrylamide)-b-polycaprolactone-b-poly (N-isopropyl acrylamide) and polycaprolactone. The release of the drug increased twice by augmentation at a temperature of only 15 °C. This system was effective in chemotherapy and hyperthermia action, achieving a synergistic anticancer effect [117].

Nagura et al. prepared micelles by self-assembling biodegradable polycaprolactone-block-poly (glutamic (PCL-b-PGA) capable of loading DOX drug and Fe_3_O_4_ nanoparticles. This system released the drug specifically into tumor cells abundant in GHS. It also presented a transverse relaxation value of 192.06 mM^−1^S^−1^. This value allows for high-quality images to be obtained in MRI even though the magnetic properties of Fe_3_O_4_ nanoparticles are influenced by polymeric coating and the hydrodynamic diameter of micelles [118].

### 3.5. Electric Field-Responsive

Micellar systems that respond to electric fields have one or more conductive polymers in the block copolymer, which cause the spontaneous orientation of their dipoles. The responsiveness is most commonly obtained using molecules that undergo an electrically induced redox reaction. These micelles show a slow drug release and low drug encapsulation, which is a challenge for future research [21,119].

Liao et al. prepared voltage-sensitive micelles composed of a copolymer (PEG-β -CD/PLA-Fc). The polymer was formed by the union of homopolymers polyethylene glycol-β-cyclodextrin (PEG-β-CD) and poly(L-lactide). When applying a positive voltage to the solution, the Fc group was oxidized to the Fc + group, subsequently abandoning the hydrophobic β-CD cavity rapidly, leading to the dismantling of the supra-molecular system (EG-β-CD/PLA-Fc). In these experiments, paclitaxel was used as a model drug. Micelles showed perfect stability until a voltage of +1V was applied, which was enough to start releasing the drug [120].

## 4. Micelle–Lipid Nanocapsules

Lipid-based nanocarriers have inspired great interest in recent times. Lipid materials allow for improved internalization in cells due to their high membrane affinity. The lipid components generally are phospholipids, cholesterol, triglycerides, bile salts, and fatty acids. Some lipids function as surfactants by forming steric barriers that prevent phenomena such as coalescence and aggregation. Positively charged lipids on the surface of nanostructures promote non-specific binding to erythrocytes, lymphocytes, and endothelial cells. In contrast, negatively charged lipids tend to decrease their potential to penetrate negatively charged cell membranes.

Lipids in nanocarriers allow for easy coating with hydrophilic polymers (PEG, Tween, PVA, etc.), notably improving their immunogenicity and circulation in the bloodstream. Lipid-based nanostructured systems that are used in drug delivery have the possibility of containing the drug within the lipid matrix, or it can be adsorbed by the polymeric coating on its surface [121]. Small-molecule hydrophobic drugs can be encapsulated in the hydrophobic core of micelles, and many water-soluble peptides and drugs are assumed to be present in the polymeric micellar layer [122].

Lipid micellar systems consist of a mixture of water or oil and surfactants. The micelles form spontaneously when the surfactant concentration reaches the CMC. These micelles are usually sterically stabilized with DSPE-PEG (PEG-conjugated 1,2-di-stearoyl-phosphatidyl-ethanolamine). On the other hand, hypothetically, the conjugation of PEG to phospholipids could give the monomers a conical morphology, facilitating their self-assembly into micelles. The size of PEG-DSPE micelles is around 7–35 nm when using small-molecular-weight polymers, whereas higher molecular weight tends to form larger micelles (Figure 5) [123,124].

Wu et al. prepared pH-sensitive poly (histidine)-PEG/DSPE-PEG copolymer micelles for cytosolic drug delivery. Poly (histidine)-PEG is too sensitive to pH, so it is necessary to combine it with DSPE-PEG so that the micelles do not destabilize prematurely at less acidic pH. Destabilization of micelles begins at pH ≈ 5.5 due to phase separation in the micelle core and dissociation of ionized PHIS-PEG molecules [125].

Na Sai et al. prepared a solution based on PEG-DSPE/solutol HS 15-mixed micelles for the ophthalmic delivery of curcumin. With these systems, it was possible to improve the solubility, stability, and permeability of the encapsulated curcumin [126].

Arias et al. designed DSPE-modified PEG-b-PCL micelles to encapsulate amphotericin B (AmB). DSPE allowed hydrodynamic diameters below 100 nm to be obtained, which features a low critical micelle concentration, high AmB-loading capacity, and little aggregation due to intermolecular interactions, such as hydrogen bonds [127].

Another good lipid system is the PCL–lipid–core nanocapsules. These vesicular structures can improve anti-inflammatory and antitumor drugs’ pharmaceutical and pharmacological aspects. Among the most used oils are caprylic/capric, triglyceride, vegetable fixed oils, octyl methoxycinnamate, mineral oil, and vitamin K1. The interfacial deposition of preformed polymer is regularly used to synthesize this type of nanocapsules. It consists of adding an organic phase that contains the hydrophobic and lipid polymers in an aqueous one containing a surfactant. The PCL lipid nanocapsules are presented in different dosage forms such as aqueous suspension, tablets, or semisolid formulations [128].

Carine et al. demonstrated the in vivo efficacy of poly (ε-caprolactone) lipid core nanocapsule loaded with acetyleugenol (AcE) to treat metastatic melanoma induced in mice. They obtained particles with hydrodynamic diameters around 200 nm using poly-sorbate 80. Other hydrophilic surfactants, such as PEG or PVA, were tested in subsequent studies. By this way, smaller nanocapsules with a lower polydispersity index were obtained [129].

Xiaodan et al. prepared nanoparticles of poly (ethylene glycol)-block-poly (ε-caprolactone) with soy phosphatidylcholine (SPC) and cholesterol for the administration of the anticancer drug paclitaxel. In this case, nanoparticles were obtained around 100 nm with a negative zeta potential value but modularly lower than −20, which can negatively affect the stability of the particles in the solution in the short or medium term. However, due to their size, the nanoparticles accumulated mainly at the tumor site, probably due to the increased permeation and retention effect (EPR) [130].

There are other types of highly stable lipid-polymeric nanoparticles with great potential in the field of controlled release of anticancer hydrophobic drugs. These nanocapsules are called cubosomes and are made up of a cubic lipid phase surrounded by an outer crown based on polymers. Cubosomes are bicontinuous liquid crystals in the cubic phase, which forms a periodic membrane structure that can accommodate water-soluble, lipid-soluble, and amphiphilic molecules. Cubosomes are more thermodynamically stable and have lower viscosity compared to other similar drug-delivery systems, such as liposomes and hexosomes. These particles will also facilitate controlled drug release due to their narrow pore sizes [131,132].

Cancer therapeutics is one of the most widespread applications of cubosomes. Prajapati et al. designed nano-self-assemblies of glycerol monooleate (GMO) sensitive to pH and capable of loading 2-hydroxyoleic acid (2OHOA). The protonation of 2OHOA in the lipid–water interfacial area determined the structural and morphological characteristics of these potential nanocarriers. When pH < 4.0, protonation of 2OHOA integrated into the hydrophobic domains of the cubosomes promoted the formation of the H_2_ phase. The increase of pH ≥ 4.0 caused the swelling of the cubic phases Pn3m and Im3m and eventually, at pH ≥ 4.5, the particles underwent a colloidal transformation from cubosomes to vesicles [133]. Chang et al. prepared curcumin-loaded cubosomes using monoolein (MO), monopalmitolein (MP), and phytantriol (PT). The encapsulation efficiency of curcumin within the cubosomes varied depending on the nanoparticle architecture and the curcumin concentration. Curcumin at low concentrations was associated with the polar region, but at high concentrations it was transferred to the region of the lipid bilayer. Due to these reasons the phytantriol cubosomes were able to encapsulate more curcumin in the lipid bilayer [134]. Pramanik et al. developed a formulation of monoolein-based cubosomes conjugated with hyaluronic acid (HA). These cubosomes belonging to the space group Im3m were loaded with copper acetylacetonate. The nanocarrier was tested in cancer cells expressing CD44, effectively killing cancer cells by inducing apoptosis [135].

## 5. Computational Approaches to Design Polymeric Micelles

One of the significant contributions of computational chemistry these days has been optimizing the time and costs of experiments in the laboratory. When drug-delivery systems are studied, the particles’ solubility parameters, size, and surface properties are analyzed, as well as the compatibility of polymeric and lipid materials with the encapsulated drug. To do so, scientists have used theoretical approaches such as the Flory–Huggins (FH) theory, analytical predictions of partition coefficients, and molecular simulations. Analytical methods such as FH help predict an early material design. FH analysis, however, presents serious errors when comparing materials of different chemical properties.

On the other hand, molecular simulations are normally carried out through the Monte Carlo method (MC) or molecular dynamic (MD) approaches. MC uses approximations by evaluating the change in potential energy, whereas MD simulations generate a trajectory of materials in time by numerically integrating the differential of the movement equations. Both simulations produce more reliable results, but at the expense of high computational costs in terms of time and technology. However, more complex systems can be analyzed more quickly and re-evaluate systems made using theoretical models [136].

Shi et al. used the Flory–Huggins interaction parameters (χFH) and polarity difference (4Xp) to evaluate drug–polymer compatibility. The authors demonstrated that the critical value of χFH = 0.5 is not appropriate to use as an evaluation criterion of drug–polymer compatibility because the theory was developed mainly for nonpolar dispersion-force interactions between solvent and polymer [137]. Chun et al. used the Flory–Huggins theory to characterize poly (2-oxazoline) molecular association based on micelles with various epoxides and diols. However, the parameter χFH had to be used in DM to be able to enrich the computational calculations [138].

Rezaeisadat et al. studied the interaction of curcumin with PNIPAAm-b-PEG copolymer nano-micelle by means of molecular-dynamics simulation. They used GROMACS as software and Amber99SB as a force field of all atoms. The results showed that the PEG group increases the density of the copolymer. It was predicted that the polymeric phase PNIPAAm-b-PEG changes at a temperature in the range of 300 to 305 K, and the number of hydration layers is reduced to a single layer, so the hydrogen bonds in the PNIPAAm-b-PEG polymer will play an important role [101,102,139].

On the other hand, Zeng et al. also used molecular dynamics to work with the 5 pol-yamidoamine-graft-poly (carboxybetaine methacrylate) system (PAMAM (G5)-PCBMA) and its ability to encapsulate DOX. The results showed that in the PAMAM (G5)-PCBMA the PAMAM dendrimer (G5) constitutes a hydrophobic nucleus to charge DOX at a physiological pH of 7.4, whereas PCBMA serves as a protective layer to improve the stability of the unimolecular core-layer micelle [140].

## 6. Current Status and Future Prospects

The design of smart polymeric micelles as anticancer nanomedicines is currently focused on partially or totally replacing chemotherapy treatments. The main objective of the research is to create a polymeric system that optimizes the physicochemical properties and pharmacokinetic/pharmacodynamic profiles of anticancer drugs approved by the Food and Drug Administration (FDA) and the European Medicines Agency (EMA). Currently, very few micellar systems have reached advanced clinical trials, since in many cases the sustained release of the drug is insufficient. On the other hand, the use of surfactants increases cytotoxicity but also improves membrane permeability, solubility, and systemic exposure. The future of polymeric micelles consists of eradicating these disadvantages caused by the excessive use of surfactants. In this sense, peptides are beginning to be designed in copolymer chains to reduce cytotoxicity and function as targeting towards specific cancer cells. In addition, it increases the sustained release of the drug through the presence of multiple functional groups in the side chains of the peptides that generate covalent bonds with the drugs.

## 7. Conclusions

In recent times, smart polymer micelles have revolutionized the field of nanocarriers. Many drugs, such as doxorubicin, paclitaxel, lapatinib, cisplatin, adriamycin, and curcumin, have been efficiently encapsulated. The results were due to the facility to change the physical and chemical properties of these nanostructures. The size, morphology, and solubility of the micelles had a significant impact on their circulation. On the other hand, a simple change in the hydrophobic or hydrophilic segments of the copolymer can generate a variation in the drug’s loading capacity and release. This release will be strongly influenced by the ability of the micelle to respond to endogenous effects (pH gradients, redox reactions, enzymes mechanisms of action) and exogenous effects (temperature, ultrasound effects, light-triggered, magnetic, electric field). The future of these micellar systems is tilted towards the formation of dual systems that combine several of these effects to optimize drug delivery. All the experimental work behind this science can be guided and optimized with the help of computational chemistry. These systems’ potential and effective application could avoid future surgery or chemotherapy risks when treating serious diseases such as cancer or chronic pain.

## Figures and Tables

**Figure 1 cancers-15-00004-f001:**
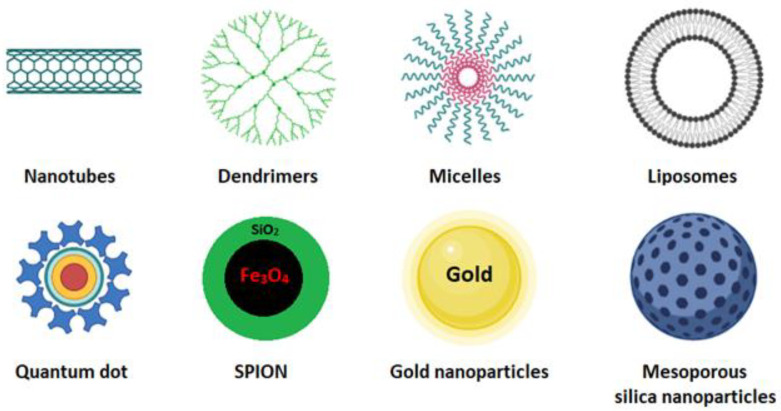
Schematic representation of some nanocarriers currently used as smart drug-delivery systems [10].

**Figure 2 cancers-15-00004-f002:**
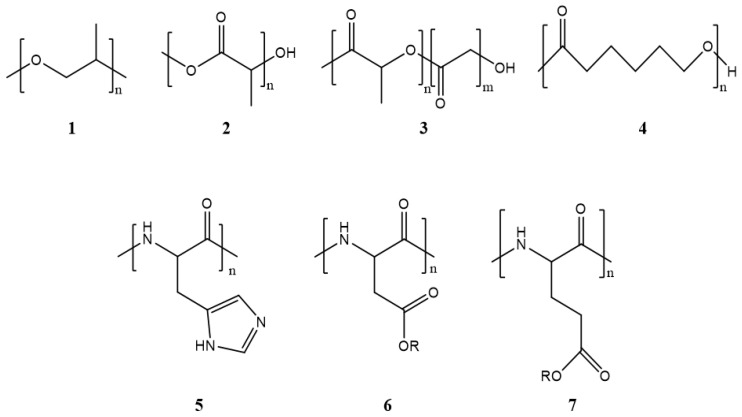
Structure of the most frequently used polymers as a core in the micelle’s preparation. Polyethers; **1**: poly(propylene oxide) (PPO) [25]. Polyesters; **2**: poly(L-lactide) (PLA) [26], **3**: poly(lactide-co-glycolide) (PLGA), **4**: poly(ε-caprolactone) (PCL) [27], **5**: poly(L-histidine)(pHis) [28], **6**: poly(L-aspartic acid) and derivatives (pAsp) [29], **7**: poly(L-glutamic acid) and derivatives (pGlu) [30].

**Figure 3 cancers-15-00004-f003:**
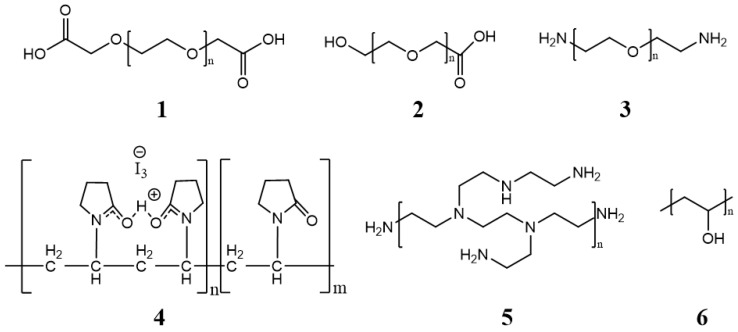
Polymers are used to coat the core of micelles. **1**,**2**,**3**: Polyethylene glycol (PEG) with functional groups at the ends (COOH, OH^−^, −NH_2_, etc.) [4,32,33]. **4**: Poli (N-vinyl-2-pyrrolidone) (PVP) [34], **5**: branched polyethyleneimine (PEI) [26], **6**: poly (vinyl alcohol) (PVA) [35].

**Figure 4 cancers-15-00004-f004:**
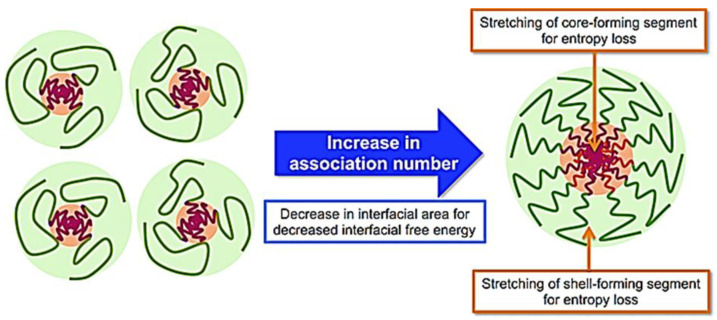
Mechanism of micelle formation from copolymers [37].

**Figure 5 cancers-15-00004-f005:**
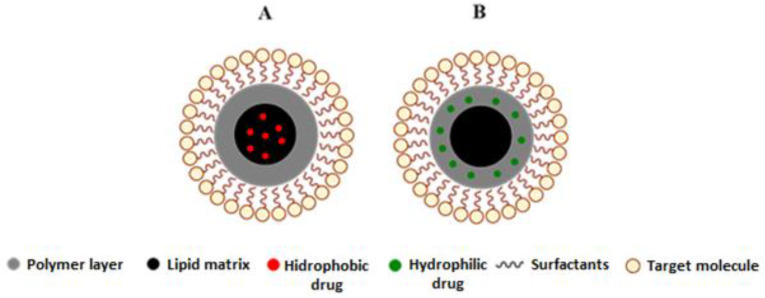
(**A**) Hydrophobic drugs encapsulated in the lipid matrix and (**B**) hydrophilic drugs adhered to the polymeric layer.

**Table 1 cancers-15-00004-t001:** Preparation methods of the polymeric micelles [31].

Methods	Advantage/Disadvantage	Drug-Loading Capacity	Solvents	Types of Drugs	Encapsulated Anticancer Drug	Polymers Used
Direct dissolution	The simplest technique to prepare polymeric micelles. Does not use organic solvents. Low-molecular-weight hydrophilic polymers	Low drug-loading capacity due to water solubility of polymers	Water	Not applicable for most hydrophobic drugs	Paclitaxel [41]	Mostly hydrophilic polymers; PLA-PEG
					Docetaxel [42]	d-a-tocopheryl PEG1000 succinate (TPGS)
					Doxorubicin [43]	Pluronic F127/poly (methyl vinyl ether-alt-maleic acid)
Oil-in-water emulsification	Easy preparation. Small particles with a narrow size distribution. Not environmentally friendly due to the use of chlorinated organic solvents.	High drug-loading capacity	Organic solvents immiscible in water (CHCl_3_, EtAc, and CH_2_Cl_2_)	Hydrophobic drugs	Doxorubicin and erlotinib [44]	PLGA/pluronic F-127
					Triptorelin [45]	PLA/PLGA
Thin-film hydration/solvent evaporation	Only applicable for copolymers with high hydrophilic–lipophilic balance (HLB). Feasible for scaling up but very expensive	High drug-loading capacity and encapsulation efficiency	Water-miscible volatile organic solvents (DMF, THF, DMSO, acetonitrile, MeOH, acetone)	Hydrophobic drugs	Doxorubicin [46]	PEG 5000-lysine-di-tocopherol succinate (P5kSSLV)
					Curcumin [47]	Poly(ethyleneoxide)-b-poly(propylene oxide)-b-poly(ethylene oxide) (PEO-b-PPO-b-PEO/pluronic F-127)
					Paclitaxel [48]	Inutec SP11 (INT)
Dialysis	For highly hydrophobic polymers with long alkyl chains. Difficulty releasing. Easy to remove organic solvents. Not applicable on a large scale due to high water consumption.	High drug-loading capacity	Water-miscible volatile organic solvents (DMF, THF, DMSO, acetonitrile, MeOH, acetone)	Hydrophobic drugs	Docetaxel [49]	PEG/hyperbranched poly(amidoamine) HAPH
					Docetaxel [50]	PLGA/PEG–maleimide
					Doxorubicin [51]	PCL-S-S- biodegradable photoluminescent polymer (BPLP)
Freeze-drying	High stability and narrow size distribution. Organic-solvent reusability. Thermolabile drug-encapsulation suitability. Limited lyophilize organic solvents and copolymers soluble in them.	High drug-loading capacity	The mixture of water and freeze-dryable organic solvents such as tert-butanol and dimethyl acetamide	Hydrophobic drugs	TM-2 [52]	mPEG/PLA
					Docetaxel [53]	Thermosensitive methoxy poly(ethylene glycol)-*b*-poly[N-(2-hydroxypropyl) methacrylamide lactate] (mPEG-bpHPMAmLacn)

**Table 2 cancers-15-00004-t002:** Selected examples of pH-sensitive chemical bonds and their degradation products [58].

Type	pH	Acid-Sensitive Chemical Bonds	Degradation Products
Vinyl ester	4.5–5.0	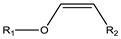	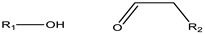
Amide	4.5–6.0	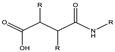	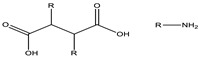
Imine	6.8	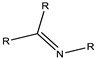	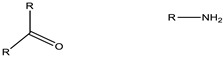
Oxime	4.8–5.0	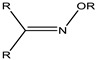	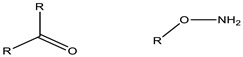
Hydrazone	5.0	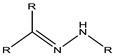	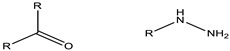
Orthoester	5.0–6.0	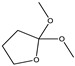	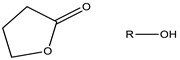

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
