# Peer review of "Smart Polymeric Micelles for Anticancer Hydrophobic Drugs"

_cancers, 2022, doi:10.3390/cancers15010004_

Round 1
Reviewer 1 Report
The authors have written a well summarized review on nanomaterials of polymeric origin used for anticancer drug encapsulation. However there is a minor suggestion for improving the review.
In line 40 of Introduction, the authors describe various class of nanomaterials used for cancer therapy and diagnosis. In this section the authors should also include ‘cubosomes’ which are an upcoming nanomaterials widely gaining medical attention. There could be a separate section on cubosomes. As cubosomes are widely used for biomedical application and could be targeted to cancers via receptors. Some supporting references for encapsulation of hydrophobic anticancer drug and imaging agents in cubosomes are (https://doi.org/10.1021/acs.molpharmaceut.2c00439).
Reviewer 2 Report
The article represents a comprehensive review on polymeric micelles, following both their obtaining and characterization (size, shape, surface charge) and their properties of being responsive to different stimulus, most of them from the perspective of the cancer environment (pH, redox potential, enzyme, temperature).
Still, I consider that two aspects should be completed/added:
i) Section 6 - Computational approaches, is presented too briefly, including in fact a review of only 5 bibliographic references, far too few in relation to the large number of studies in the field. Therefore, I suggest either removing this section, or completing it with more results, possibly summarized similar to the ones from the other paragraphs, in the form of a table, including information such as the software used, calculated parameters, as well as if (or how) the simulation results were validated, with emphasis on the parameters related to cancer environment.
ii) I suggest a to include also a final section regarding the current status of the use of polymeric micelles in the therapeutic treatment of cancer (approved and actually used), as well as future perspectives in this direction.
Reviewer 3 Report
The submitted review describes the recent achievements in designing new systems for drug delivery (DDS). The effective and correct delivery of required drug to biotarget is crucial for the therapeutical success. Authors describe the basic factors deciding of stability of micelles in the human body and characterize the types of stimuli necessary for drug release and response of micelles material. The concise description of computer aided polymer micelles designing enriches the review. The topic of paper is actual and presents modern polymeric materials use in formation of micelles, which are the most effective carriers for drugs.
The text needs several improvements before acceptation for publication.
First, authors are asking for information about molecular mass of described polymeric materials.
Carefully text examination is necessary for abbreviation which are not explain in text
Table 1 last column: PEO, HAPH, BPLP, mPEG-bpHP-MAmLacn
Page 7 line 216: MEO2MA-co-OEGMA
Page 8 line 249; poly (beta-aminoester) (PAE) what acid it contains?
Page 8 line 253: poly (DEA) is equal to PDEA?
Page 9 line 266: 3,3'-diselanodioldipropanoic acid (DSeDPA) it means 3,3'-Diselenodipropionic acid?
Page 9 line 269: TPE-SS-PLAsp-b-PMPC what is a composition of this copolymer
Page 9 line 286: PEG-PBLA-CC-PTX (PPCP)?
Page 11 line 418: what means polypeptide nanoconjugates UPDN?
Page 12 line 422 PEO?
Page 12 line 43: added diphenylodonium-2-carboxylate (PAG) should be diphenyliodonium-2-carboxylate?
Page 12 line 468 what means cypat? It should be cypate (as a carbocyanine dye) or something else?
Page 13 line 518: what means PLLA?
Page 13 line 519: is poly(L-lactide) in page 3 line 88: is PLA, please use the same abbreviation.
Page 14 line 543: is dis-tearoyl-phosphatidylethanolamine, please change to di-stearoyl-phosphatidyl- ethanolamine
I suggest introduction of simplified structures for discussed copolymers
Page 8 line 235 is an acid pH should be acidic pH
Table 2 column 1 is Hidrazone, should be Hydrazone
The references are actual, I suggest consideration of following papers:
Akihiko Kikuchi, Teruo Okano, Advanced Drug Delivery Reviews 54 (2002) 53–77.
Ghanashyam Acharya, Kinam Park, Advanced Drug Delivery Reviews 58 (2006) 387– 401. doi: 10.1016/j.addr.2006.01.016
Jinhyun Hannah Lee, Yoon Yeo, Chem Eng Sci. 2015, 125: 75–84. doi: 10.1016/j.ces.2014.08.046.
In references list please complete the following positions: 16, 17, 21, 40, 41, 43, 44, 45, 46, 47, 48, 49, 51, 52, 57, 58, 59, 60, 88, 89, 96, 97, 106, 116, 117.
